# Crosstalk between Hepatitis B Virus and the 3D Genome Structure

**DOI:** 10.3390/v14020445

**Published:** 2022-02-21

**Authors:** João Diogo Dias, Nazim Sarica, Axel Cournac, Romain Koszul, Christine Neuveut

**Affiliations:** 1Laboratoire de Virologie Moléculaire, Institut de Génétique Humaine, CNRS, Université de Montpellier, 34000 Montpellier, France; joao.dias@igh.cnrs.fr (J.D.D.); nazim.sarica@igh.cnrs.fr (N.S.); 2Unité Régulation Spatiale des Génomes, CNRS, UMR 3525, Institut Pasteur, Université de Paris, 75015 Paris, France; axel.cournac@pasteur.fr (A.C.); romain.koszul@pasteur.fr (R.K.)

**Keywords:** HBV, HBV cccDNA, spatial nuclear organization, gene regulation, DNA viruses

## Abstract

Viruses that transcribe their DNA within the nucleus have to adapt to the existing cellular mechanisms that govern transcriptional regulation. Recent technological breakthroughs have highlighted the highly hierarchical organization of the cellular genome and its role in the regulation of gene expression. This review provides an updated overview on the current knowledge on how the hepatitis B virus interacts with the cellular 3D genome and its consequences on viral and cellular gene expression. We also briefly discuss the strategies developed by other DNA viruses to co-opt and sometimes subvert cellular genome spatial organization.

## 1. Introduction

The hepatitis B virus (HBV) is a widespread pathogen and HBV infection is one of the major risk factors for hepatocellular carcinoma (HCC). Despite the development of an effective HBV vaccine, the number of chronic HBV-carriers has reached 257 million individuals worldwide, of whom it is estimated that ~25% will die of liver cirrhosis or HCC [1].

HBV is a small (3.2 kb) enveloped DNA virus that replicates its genome in the cytoplasm by reverse transcription of encapsidated pre-genomic RNA (pgRNA) into a partially double-stranded relaxed circular DNA (RC-DNA). Upon internalization and import into the nucleus, the RC-DNA is repaired and converted into covalently closed circular DNA (cccDNA) that serves as a template for all HBV transcripts, including pgRNA [2,3]. Viral replication relies exclusively on the maintenance of episomal cccDNA, but the integration of HBV DNA into the host genome is a frequent event and occurs by mechanisms that are still unknown [4,5]. HBV cccDNA is organized into a chromatin-like structure, associated with histone and non-histone proteins [6,7,8,9]. In this context, HBV RNA expression is controlled by 4 promoters: the Pre-core/core, Pre-S1, PreS2/S and the HBx promoters, and two enhancers. These transcription regulatory sequences contain binding sites for both ubiquitous and liver-specific host factors [10]. HBV cccDNA also contains three CpG islands that are believed to control HBV transcription [11,12]. Besides host factors, HBV transcription requires the regulatory protein HBx that is essential for viral replication in vivo. HBx has been shown to be needed for the establishment and maintenance of an active chromatin state [12,13,14]. HBx transcriptional activity relies first on its ability to counteract the repressive activity of the structural maintenance of chromosome 5/6 (SMC5/6) by triggering its degradation via its interaction with the E3 ubiquitin ligase CulA/DDB1 [15]. HBx is also recruited on the cccDNA and may modulate chromatin state and transcription through the interaction with transcription factors and chromatin-modifying factors [8,16].

HBV episomal DNA establishment and maintenance, or HBV linear DNA integration and expression of HBV genes, do not occur randomly, nor in a “neutral” chromatin context, and may have to adapt to the existing cellular mechanisms that govern transcriptional regulation. While it is well documented that viruses that transcribe their DNA in the nucleus (including HBV) modify or repurpose the host transcriptional machinery to induce viral gene expression, new questions have emerged about the mechanisms of viral transcriptional regulation in light of novel insights into the spatial organization of the cellular genome and on the close relationship between 3D genome organization and genome function [17,18,19,20].

Groundbreaking discoveries in 3D genome organization and gene expression regulation have been made possible by the improvement and development of imaging techniques such as fluorescence in situ hybridization (FISH) and super resolution microcopy and biochemical and genomic approaches, such as chromosome conformation capture (Hi-C) methods [21,22,23,24]. Studies by several groups have shown that chromosomes exhibit a hierarchical, non-random organization in the nucleus at all stages of the cell cycle, in agreement with the earlier observations, showing that they occupy specific regions in the interphase nucleus (chromosome territories), with chromosome intermingling found at the edges of territories (Figure 1) [25].

Another level of organization revealed by Hi-C studies is the partition of the genome into two compartments: compartment A, which contains the gene-rich and active euchromatin regions, and compartment B, which contains the gene-poor, repressed heterochromatin regions [24]. A and B compartments are frequently associated with different nuclear areas, with the B compartment locating to the periphery of the nucleus or at chromocenters, while the A compartment occupies a more central position in the nucleus [18,26,27]. Compartment A and B content in humans is, to some extent, cell type-specific, with ~60% of regions able to switch between the two during differentiation, as revealed by the analysis of 21 different cell types [28]. On a smaller scale, ~90% of the genome is segmented into sub-megabase, topologically associating domains (TADs; ~100 kb to 1 Mb) defined as chromosomal DNA regions that display increased contacts within themselves rather than with neighboring regions [29,30]. TADs are insulated from each other by conserved boundaries, often associated with the structural maintenance of chromosome (SMC) complex cohesin and the zinc-finger protein CCCTC-binding factor (CTCF) [17,18,31,32]. TADs and their boundaries tend to be conserved across cell types, with genes inside a TAD often displaying similar expression patterns constituting insulated units of coregulated genes [28,29,30,32,33]. Inside TADs, sub-networks of contacts (such as promoter-enhancer looping) appear to correlate with different expression levels and epigenetic states, and to vary upon differentiation with the establishment of contacts involving lineage-specific enhancers and transcription factors [34,35]. Moreover, single cell data using high resolution microscopy favors a model where contacts and intermingling mostly occur dynamically within a TAD region, and chromatin nanodomains form an additional layer of chromatin organization at a sub-TAD level [36]. Disturbing insulation between neighboring chromosome domains has important implications for gene regulation, with enhancers gaining or losing access to promoters and genes, which can disturb the normal regulation of gene expression in development and disease [37,38,39,40,41,42,43]. As mentioned previously, chromatin is partitioned based on a transcriptional state forming separate compartments where regions with similar epigenetic profiles tend to contact each other. However, in addition to intra-chromosomal contacts, chromatin is also engaged in inter-chromosomal contacts that may promote the transcription of co-regulated genes or RNA processing [44,45,46,47,48]. Spatial clustering of different genomic regions in membrane-less condensates can occur by dynamic interactions between DNA, RNA and proteins at higher localized concentrations. This process can be explained by liquid–liquid phase separation (LLPS), a concept originally from theoretical physics, proposing that certain chromatin-associated proteins in solution, above a critical concentration, form separate condensates in a reversible manner though weak and multivalent interactions. These condensates can be viewed as droplets with specialized micro-environments that can regulate specific biochemical processes. This process is often illustrated by the formation of oil droplets in an aqueous solution. The ability to form condensates is influenced by the presence, in proteins, of low-complexity domains or intrinsically disordered regions (IDRs), polypeptide sequences that lack a stable defined structure. IDRs in histones, transcription factors, RNA polymerase II and other chromatin-associated proteins mediate interactions between proteins, DNA and RNA and are thought to play an important role in chromatin condensate formation [49,50,51]. Thus, a more complex model for the regulation of gene expression is emerging which relies on the spatial organization of biochemical complexes in membraneless structures, including nuclear bodies. Nuclear bodies, structures visible by microscopy, are composed mostly of proteins and RNAs that dynamically associate and dissociate and that regulate specific nuclear processes and contribute to 3D genome organization. Some nuclear bodies are directly linked, or result from transcriptional activity: the nucleoli, where ribosomal genes are transcribed and the pre-ribosomal RNAs processed; the nuclear speckles, structures enriched in splicing factors and snRNPs, associated with transcription, RNA processing and RNA export; and transcription factories, 50–180 nm nuclear foci enriched for active RNA polymerases II associated with nascent RNA synthesis and components of the transcription and RNA-processing machineries [44,50,52,53,54]. Transcriptional hubs may provide specialized microenvironments favoring a more efficient transcription and may even promote the recruitment of genes to these specialized areas, as has been suggested for genes containing binding sites for CTCF [55], KLF1 [48] or NF-kB [56]. Several studies identified the clustering of co-regulated genes that cooperate with each other to enhance gene expression [57,58]. Furthermore, it has been shown that some genes in the cluster can influence the transcription of the others, highlighting the importance of this 3D organization in transcriptional regulation [46,58]. Recently, new tools and new methods, namely split-pool recognition of interaction by tag extension (SPRITE), genome architecture Mapping (GAM) or tyramide signal amplification sequencing (TSA-Seq), have been developed to capture these complex DNA contacts that occur simultaneously within the nucleus and to study their arrangement around nuclear bodies [26,59,60,61]. These studies reveal that chromatin is engaged in multiple higher-order contacts inside hubs that organize from nuclear structures. Highly transcribed gene-rich regions organize around nuclear speckles or regions that can be described as transcription factories, while gene-poor and transcriptionally inactive regions are organized around nucleolus or nuclear lamina [26,61].

The cellular genome is thus subject to a highly hierarchical organization that regulates gene expression. Recent studies suggest that viruses are no exception and viral gene regulation depends on viral genome positioning that occurs in the context of the 3D structure of the host genome. In this review, we will summarize the current understanding of how HBV co-opts cellular genome compartmentalization to transcribe cccDNA. We will briefly discuss strategies developed by other DNA viruses that transcribe their genome in the nucleus.

## 2. Interaction between Episomal Viral Genomes and Cellular Genome from a 3D Perspective

### 2.1. HBV

In a recent study, we addressed the question of whether non-integrated viral DNAs distribute randomly in the nucleus, or target specific regions in the higher-order architecture of mammalian genomes [62]. Two DNA viruses, hepatitis B virus (HBV) and human adenovirus type 5 (Ad5), were tracked in either primary human hepatocytes (PHH) infected in vitro by HBV or PHH, derived from a 7-month-old donor infected by the human adenovirus serotype 5 (Ad5). The interplay between these two DNA viruses and the host genome from the viewpoint of their genome organization was investigated using Hi-C, showing that infection with HBV or Ad5 does not induce large-scale reorganization of the hepatocyte genome.

To enrich for HBV/cellular DNA contacts, a capture approach (CHi-C), using biotinylated oligonucleotides covering the viral genome, was applied, showing that the HBV genome makes contact with the entire host genome, with a significant preference for active chromatin regions of compartment A enriched in active histone marks (H3K4me3, H3K4me2 as well as H3K27ac). Further analysis has demonstrated that HBV contacts preferentially CpG islands (CGIs). These regions, present in eukaryotic promoters, are mainly unmethylated and control transcription initiation through the establishment of an active chromatin state [63]. Unmethylated CGIs are recognized by proteins containing Zinc Finger (ZF)-CxxC domains, such as Cfp1 (CXXC finger protein 1), which in turn recruit the methyltransferase Set1 responsible for H3K4me3 deposition [63]. Using siRNA and chromatin immunoprecipitation (ChIP) experiments, we showed that Cfp1 is recruited to HBV cccDNA and is required for H3K4me3 deposition and HBV transcription. These results suggest that HBV preferentially contacts regions in the host genome that are enriched in factors important for its own transcription, such as Cfp1. Finally, we generated RNA-seq data from PHH infected by HBV, and we found that contacts were significantly enriched at CGIs associated with highly expressed genes and genes deregulated during infection, suggesting that HBV contacts may interfere with cellular gene expression.

Regarding Ad5, we observed that while viral DNA also contacts preferentially active regions, contacts are enriched at TSS and enhancers of highly expressed genes, but also genes up-regulated during the infection, suggesting again that contacts may interfere with cellular gene expression. Regions contacted by Ad5 are enriched for motifs such as FOXO A1 and FOXO A2 and CAAT enhancer binding protein (C/EBP). While it is not known whether FOXA factors are involved in Ad5 transcription, they do, however, interact with factors such as the mediator, which is involved in Ad5 expression. C/EBP has been shown to regulate Ad5 major late promoter transcription. Altogether, our data suggest that DNA viruses infiltrate the 3D genome and target specific active regions that may provide an environment favorable for their own transcription/replication. Further studies will be needed to determine the mechanism driving the localization of viral DNA at these discrete locations.

Other groups, using different approaches, have explored the spatial organization of HBV and have reached a similar conclusion that HBV does not randomly localize in the nucleus, but contacts favorable active regions. Yang and collaborators used high-throughput wide translocation sequencing coupled with chromosome conformation capture (HTGTS-3C) to study both HBV cccDNA and integrated HBV DNA 3D organization [64]. This approach, based on 3C and linear amplification-mediated high-throughput genome-wide translocation sequencing (LAM-HTGTS), allows the identification at high-resolution of contacts occurring between a bait (here HBV DNA) and unknown DNA sequences [65]. Using hepG2-NTCP-infected cells and HepaD38 cells containing, depending on the culture conditions, both HBV-integrated sequences and HBV cccDNA, they showed that HBV cccDNA contacts preferentially active cellular DNA. Their results confirm that HBV cccDNA contacts TSS and regions enriched for CpG islands. Moreover, they observed an increase in contacts at enhancers characterized by H3K4me1 deposition. Interestingly, the authors observed an increase in contacts on chromosome 16, 17, 19, 20. However, the significance of this finding was not discussed and will require further investigation. It will be interesting to determine whether these chromosomes contain more genes that are highly expressed in hepatocytes, or genes involved in regulatory communities enriched for factors involved in HBV regulation.

Contacts between HBV episomal DNA and cellular enhancers has also been observed using a 3C approach [66]. In this work, the authors identified a preferential interaction between HBV and the chromosome 19p13.11 region, which contains a highly active enhancer. Using cell lines stably knockout for this enhancer, the authors showed that HBV expression was decreased, suggesting that enhancer 19p13.11 indeed positively impacts HBV transcription. However, they did not assess whether the deletion of the enhancer modified HBV/19p13.11 DNA contacts. To address this point, the authors rather used engineered DNA binding molecule mediated chromatin immunoprecipitation (enChIP) to study the role of YY1 protein that binds both the 19p13.11 enhancer and HBV cccDNA and that has been shown to be involved in chromatin loop formation and genome organization [67,68,69]. They showed that YY1 is co-immunoprecipitated with both the HBV DNA and the 19p13.11 enhancer, while silencing of YY1 or mutation of YY1 binding site in HBV genome decreases the co-immunoprecipitation of the 19p13.11 enhancer and YY1. While this result suggests that YY1 is involved in the HBV cccDNA/19p13.11 enhancer contacts, further analysis will be needed to strengthen this finding and also to investigate whether YY1 is important for the establishment of HBV contacts with other host genomic regions.

Finally, in a recent work, Tang and collaborators analyzed the spatial position of the HBV genome, which is deficient for HBx expression and was thus transcriptionally silenced using 4C analysis in HepG2 NTCP and PHH-infected cells [70]. They observed that HBV X- genome contacts preferentially chromosome 19 at repressed regions marked by H3K9me3 (albeit less frequently, contacts were also observed with chromosomes 1, 16, 17, 20 and 22). A similar analysis of episomal HBV wt DNA distribution revealed that episomal DNA contacts preferentially chromosome 19, 22 and 17, with an enrichment for chromosome 19. In this context, cellular DNA regions contacted by HBV DNA are enriched for active marks. Their findings were confirmed at the single cell level using in situ immunofluorescence hybridization (FISH). Finally, they observed that a re-expression of HBx or silencing of *smc6*, involved in the transcriptional repression of HBV X-, leads to the relocation of HBV DNA to active regions on chromosome 19. Altogether, these studies show that HBV DNA is not randomly localized in the nucleus. Active viral DNA contacts preferentially active chromatin, while transcriptionally repressed DNA is associated with repressed chromatin and contacts preferentially the B compartment. Some of the differences found in the aforementioned studies may result from using infection systems versus transfection, different cell lines (transformed hepatoma cells or primary human hepatocytes) leading to a different number of cccDNA molecules per cell, which can have an impact on the detection of viral/host contacts. Moreover, chromosome 19, found to be preferentially contacted in some of the studies, has some unique features: it is located in a central position in the nucleus, it has the highest gene density of all the chromosomes, a high density of repetitive sequences (particularly Alu repeats) and the highest GC content of all chromosomes with 2/3 of the genes having at least one CpG island [71,72].

Of note is the fact that HBV X- cccDNA relocates at active regions in cells silenced for *smc6*, which suggests that HBx per se is not directly promoting the targeting of HBV DNA to active regions, as suggested by Hensel et al. [73], but rather the transcriptional status of cccDNA may impact on its localization. Localization at active chromatin may be driven by cellular factors, epigenetic modifications or viral factors. Interestingly, the SMC5/6 complex has been shown to be responsible for HBV transcriptional repression acting as a restriction factor for HBV, and more recently, has been found also to restrict transcription from unintegrated HIV-1 [74]. SMC5/6 belongs to the structural maintenance of the chromosome (SMC) family of proteins, together with cohesin and condensin. SMC proteins form ring-like complexes that can bind, compact and extrude DNA-forming loops in an ATP-dependent reaction [75,76]. Condensin and cohesin have multiple roles in genome organization during interphase and mitosis and are fundamental to the structural maintenance of chromosomes. Cohesin holds sister chromatids together and is a key factor in the folding of the chromosome into loops, while condensin is involved in chromosome compaction during cell division [76]. The SMC5/6 complex has been previously implicated in DNA repair and DNA replication. More recently, two studies using biochemical and biophysical approaches have shown that the SMC5/6 complex can bind and compact unusual DNA structures [77,78].

The mechanism of cccDNA repression by the SMC5/6 complex remains largely unknown and one can envision several possibilities. The SMC5/6 complex binds to the HBV genome due to its episomal nature and is responsible for targeting it to repressed regions, then promoting its transcriptional silencing. Alternatively, the binding of SMC5/6 contributes to the deposition on HBV X- cccDNA of H3K9me3 and HP1, which in turn are responsible for the subsequent repositioning at repressed regions [12,51]. Another possibility is that compaction and conformational changes as well as the repressed chromatin state associated with SMC5/6 might prevent the association with cccDNA of host factors that mediate its preferential localization to active chromatin regions.

In conclusion, while the different studies agree that HBV DNA genome is not localized randomly in the nucleus but contacts active regions, the mechanism responsible for cccDNA nuclear localization remains unknown (Figure 2). YY1 has been suggested to be involved in the establishment of contacts between HBV and enhancer 19p13.11, but this awaits further investigation. We and others observed that the HBV genome preferentially contacts CpG-rich regions that have been proposed to contribute to spatial segregation of active genomic regions [55]. Since the HBV genome carries 3 CGIs, it is possible that CGIs contribute to the recruitment of cccDNA molecules to the host genome. Spatial segregation of cccDNA may thus be promoted by the existence of nuclear compartments, e.g., nuclear speckles, nucleolus, polycomb domains or transcription factories enriched for specific regulatory factors that are believed to contribute to three-dimensional genome organization [26,44,53,60,61]. It has been shown that co-regulated genes colocalization may be established and/or maintained by transcription factors such as KLF1, NF-kB but also chromatin bridging proteins, such as CTCF [44,48,55,56,57]. It was recently shown that CTCF and YY1 can bind to the HBV genome as well as several other transcription factors that are recruited to HBV promoters and enhancers and that could be involved in the localization of HBV DNA to active sites of transcription [10,79]. Whether or not HBV cccDNA nuclear localization is mediated by cellular factors will need further investigation. Additionally, the epigenetic state of chromatinized cccDNA, i.e., either repressive chromatin associated with trimethylated lysine 9 of histone H3 (H3K9me3) and heterochromatin protein 1 (HP1), or on the contrary active acetylated chromatin, could be associated with different protein complexes and RNAs and could contribute to the organization and regulation of the HBV genome through liquid–liquid phase separation [44,49,80]. Alternatively, translocation at active chromatin may be mediated by a viral factor. However, as mentioned previously, HBV X- cccDNA can be located at active chromatin, suggesting that viral factors other than HBx are involved in cccDNA spatial organization. The capsid protein HBc has been shown to be recruited on both viral cccDNA and cellular promoters, which correlate with the modulation of their transcriptional activity [7,11,12,81,82]. We do not yet fully know the role of HBc on the cccDNA regulation; specifically, its contribution in cccDNA transcriptional regulation remains debated, but it is well established that HBc binds to cccDNA, modulating its structure [6,7,11,83]. Initial studies have shown that HBc is one of the non-histone proteins associated with chromatinized cccDNA and its binding to cccDNA is linked to shorter nucleosome spacing, and it is enriched at HBV CpG islands [6,7,11]. A recent publication has shown that HBc, present in the incoming viral particles and delivered into the nucleus, is recruited onto the cccDNA shortly after infection, suggesting that HBc can contribute to the establishment of viral DNA positioning before viral genes are expressed [84].

### 2.2. Other DNA Viruses

Recent studies have starting to address the role of 3D chromatin organization in viral replication and transcription. The viral protein NS1 of the parvovirus minute virus of mice (MVM) has been shown to be involved in the localization of MVM genomes to cellular DNA damage sites that occur naturally in cells as they progress through the S-phase and that are required for the establishment of viral replication [85,86]. Interestingly, since NS1 is not present in the early steps of infection, other proteins such as the viral protein of the capsid or the cellular proteins CTCF or MDC1, may be involved if localization of MVM at DNA damage sites occurs at the earliest stage of infection. Epstein–Barr virus episome (EBV) also establishes contact with the cellular genome, but the regions seem to differ according to the cell type and to the viral replication state (latency or reactivation). In the Epstein–Barr virus (EBV), positive Burkitt’s lymphoma cell lines, latent viral episomes contact cellular genomic regions enriched for binding sites for the viral protein EBNA1 and for B-cell factors EBF1 and RBP-jk, suggesting that both EBNA1 and cellular factors may contribute to EBV genome docking [87]. In these cells, cellular genomic sites contacted by EBV are marked by H3K9me3 and are flanked by AT-rich DNA and contain genes that are repressed. The role of EBNA1 in the positioning of EBV at specific regions of the host genome is further supported by the fact that silencing of EBNA1 is corelated with the loss of contact between the EBV genome and the host chromatin and the up-regulation of contacted genes. It remains, however, unclear whether these contacts are required for EBV silencing [87]. Moquin and colleagues studied the interaction between EBV and the host chromosome during latency and reactivation using the Akata-Zta cell lines containing the EBV genome that can be reactivated upon the expression of the viral protein BZLF1. They thus show that EBV episome localization varies from repressive human heterochromatin to the euchromatin region during reactivation. Again, whether this relocation of EBV at euchromatin impacts on EBV gene regulation is not determined, nor is the mechanism leading to interactions with active chromatin regions [88].

Altogether, these studies highlight the fact that the DNA virus co-opts the rules pertaining to the highly hierarchical organization of the cellular genome for their transcription/replication. They infiltrate the 3D genome using different strategies and contact specific regions that likely provide an environment favorable to their own transcription/replication. In doing so, viral DNA/ host DNA contacts in turn may impact on the gene regulation of cellular genes. Indeed, we observed that CGIs contacted by HBV are often associated with highly expressed genes and genes deregulated during infection, suggesting that viral DNA contacts could interfere with cellular gene expression. Similar results were reported by Yang and collaborators, who observed that host DNA regions contacted by HBV are enriched for genes deregulated by HBV infection [64]. Gene expression changes for host genes contacted by viral genomes is likely a common feature of infection by DNA viruses, since we observed that genes contacted by Ad5 are up-regulated in Ad5-infected arrested human fibroblasts, suggesting a causal relationship between viral contacts and gene expression changes. Recently, Okabe and collaborators have shown that in EBV-positive gastric cancer cell lines (GC), the episomal viral genome alters a host epigenome landscape, inducing heterochromatin-to-euchromatin transitions at EBV-interacting host genome loci. This mechanism, named “enhancer infestation”, is responsible for the epigenetic reprograming of host enhancers and the subsequent activation of cancer-related genes [89]. Of note, this mechanism may be cell-specific and/or dependent on the type of infection (latent or lytic), since Kim et al. observed that in BL cells, host genes bound by EBNA-1 and contacted by EBV genome are silenced, marked by H3K9me3 [87]. It is not known how HBV/host DNA contacts impact on cellular gene expression. As mentioned before, studies have shown that in mammalian systems, co-regulated genes form clusters and associate with foci enriched for RNA polymerase II and specific regulatory factors [57,60]. In these clusters, genes may cooperate with each other to enhance gene expression [46,48,56,58,90]. Moreover, the transcription of some genes in the cluster seems to be required for the transcription of the others, highlighting the importance of this 3D organization in transcriptional regulation [46,58]. The mechanisms leading to clustering and to the transcriptional cooperation are not yet understood. Clusters may favor the establishment of transcriptional areas with a high local concentration of factors regulating transcription and creating thus a specialized microenvironment allowing efficient transcription [57,60,90]. In our study, we observed, however, that HBV contacts are enriched at CGIs associated with genes that are either up- but also down-regulated upon HBV infection. It is possible that HBV DNA could disrupt the spatial clustering of CGIs, leading to the down regulation of some cellular genes, or could compete for factors required for the host gene transcription.

## 3. Interaction between Integrated HBV Genomes and the Cellular 3D Genome

As mentioned in the introduction, HBV integration is a frequent event that occurs early in infection [4,5]. The integrated HBV genome is replication-incompetent, but is thought to contribute to the development of liver cancer in multiple ways [1]. Integrated HBV sequences have been found in most HBV-related HCCs (70–90% of HCC) and are usually clonal or oligo-clonal [91,92,93]. These integrated sequences may be responsible for insertional mutagenesis and/or genomic rearrangements. However, only a few studies have aimed to properly decipher the mechanism responsible for the deregulation of cellular genes upon HBV integration. It has been shown that the integration of HBV in the *TERT* promoter leads directly to its up-regulation [93,94]. Some integrations can, however, be located far away from the targeted gene, such as the HBV sequence located 25 kb away from *TERT*. In such circumstances, cellular gene deregulation may thus require an alternative mechanism, such as long-range interactions [95]. Alternatively, HBV integration can also disrupt 3D genome organization. Indeed, HBV integration has been shown to be associated with deletions of variable lengths, as well as rearrangements at the viral-cellular junction [96,97,98]. Copy number variations (deletion and amplification), as well as translocations and inversions, are common genome aberrations involved in 3D architecture disruption and the deregulation of gene expression in cancer cells [38,42,99]. HBV integration may therefore be responsible for 3D disruption. Increasing evidence has also suggest that viral DNA insertion into the cellular genome can impact on 3D cellular organization. In mice, murine leukemia virus (MLV) can deregulate host transcription by acting both on genes located in close proximity to the integration site, and also on distant cellular genes, suggesting a mechanism involving long-range interactions [100]. These observations are further supported by another study showing that MLV can cause abnormal host transcription in tumors through integration in chromosomal regions involved in 3D interactions and enriched in cancer genes [101]. Similarly, integration of the human leukemia virus type 1 (HTLV-1) alters the genome structure by forming loops between the viral LTR and the host genome located both in the proximity and up to 300 kb away from the viral integration site [102]. Recent studies on human papilloma virus (HPV) have revealed that in cervical cancer, short- and long-range contacts between integrated HPV genomes and host chromosomes can disrupt normal TAD organization and lead to gene expression changes [103]. Chromatin surrounding HPV integration sites was found to have increased accessibility, which was associated with CTCF binding to the integrated HPV DNA and was correlated with changes in transcription and alternative splicing of neighboring genes [104]. A long-range impact of HPV integration has been demonstrated in the context of MYC deregulation [105]. Interestingly, in at least one studied case, HPV integration disrupted the local chromatin folding, resulting in the subdivision of a domain into two smaller ones. This local alteration of the chromosome architecture led to the deregulation of two cancer-associated genes [106].

The HBV genome contains binding sites for CTCF, YY1 and other transcription factors that, upon integration, can potentially disturb the local chromatin organization; for instance, this includes the formation of de novo loops. As a result, HBV regulatory regions could be repositioned in the vicinity of distal genes. Although the HBV genome is rather small compared to other viruses, the presence of regulatory elements (4 promoters and 2 enhancers) and the aforementioned binding sites for TFs known to organize the genome may impact on the proper folding of the host genome and deregulate the genes important for cancer development.

## 4. Conclusions

HBV, like other viral pathogens, exploits the 3D genome organization for its own transcription (Figure 2). However, further studies are needed to understand how the virus positions itself non-randomly at these chromatin sites and whether the disruption of these spatial localizations can lead to its silencing and even elimination. Additionally, HBV not only exploits genome folding for its own transcription, but we have observed that viral DNA contacts interfere with cellular gene expression at least in primary human hepatocytes infected by HBV. While this finding needs to be confirmed in vivo, it may contribute to cellular transformation, as suggested for EBV [89]. Finally, we and others have observed that a transcriptionally silenced virus, e.g., HBV X-, still establishes contact with cellular chromatin [62,70]. Although the regions contacted by HBV differs between the two studies, which may be due to number of cccDNA molecules per cell that may modify the distribution of HBV/host DNA contacts, it will nevertheless be important to assess whether a transcriptionally silenced cccDNA can impact cellular gene expression and epigenetic regulation through DNA/DNA contacts. Therapeutic approaches for chronic carriers aimed at silencing HBV cccDNA may therefore not eliminate the entire risk of developing HCC if silenced cccDNA molecules still establish contacts with host chromatin, which could lead to the deregulation of genes relevant to cancer development.

Finally, as viral DNA integration occurs in a context where gene expression and genome 3D organization are tightly linked and influence each other, HBV-induced HCC should be explored in the context of the 3D genome organization. Investigation of 3D genome disruption in cancer is gaining momentum and studying how DNA viruses alter the 3D genome organization will contribute to a better picture of how 3D structure deregulation can contribute to tumorigenesis. Moreover, deciphering the molecular mechanisms involved in liver cell transformation following HBV integration may bring useful insights into the rational development of targeted therapeutic agents.

## Figures and Tables

**Figure 1 viruses-14-00445-f001:**
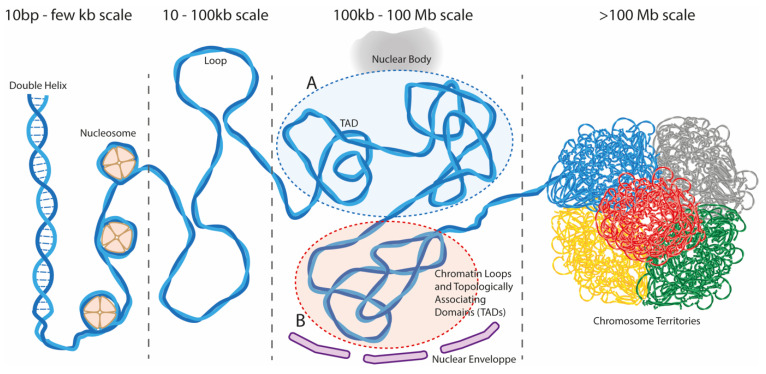
Schematic representation of the eukaryotic genome. Cellular chromosomes are organized into hierarchical domains at different genomic scales. DNA is wrapped around nucleosomes composed of an octamer of core histones, forming the chromatin fiber. Chromatin folds and forms chromatin nanodomains and loops of 10 to 100 kb in size. Loops often support enhancer–promotor contacts that control gene expression. At a higher scale from 100 kb to a few megabases, DNA loops and nanodomains are organized into topologically associating domains (TADs) corresponding to partially insulated domains characterized by preferential contacts within themselves rather than with neighboring regions. At a higher scale, up to 100 megabases, DNA fibers separate into hubs of active (A compartment) and inactive (B compartment) chromatin, clustering around nuclear bodies such as nucleolus, nuclear speckles, and transcription factories. Of note, in addition to intrachromosomic contacts, hubs have formed around nuclear bodies contain interchromosomic contacts delineating chromatin regions sharing a common function. At the highest topological level, individual chromosomes (represented in different colors) tend to occupy a distinct volume in the nucleus, defined as chromosome territories.

**Figure 2 viruses-14-00445-f002:**
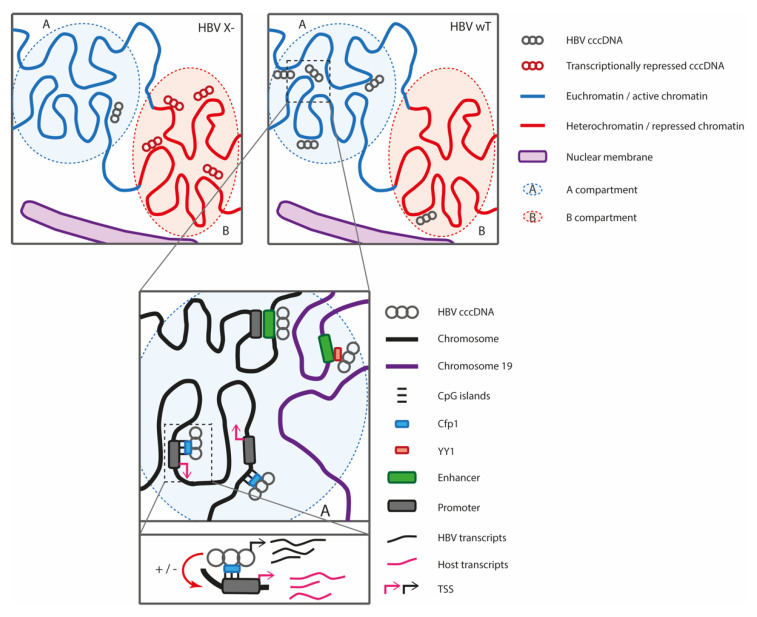
Graphical summary of HBV interactions with 3D genome. HBV cccDNA in cells infected with a wild-type virus (HBVwt) is preferentially associated with active chromatin and compartment A genomic regions (top right panel). HBV, deficient for the expression of transcriptional regulator HBx (HBV x-), is transcriptionally repressed and is associated with SMC5/6 complex and interacts preferentially with heterochromatin and compartment B genomic regions (top left panel). A zoomed version of HBV wild-type infection (middle panel) highlights the preferential contacts of cccDNA with CpG islands enriched for Cfp1, transcription start sites (TSS) and enhancers. HBV cccDNA is associated with chromosome 19 and contacts an active enhancer. This spatial association is mediated by YY1 and HBx. Lower panel: the non-random location of HBV and association with euchromatin provides a favorable environment for HBV transcription. HBV contacts in turn interfere with cellular gene regulation and can lead to up or downregulation of host genes.

## Data Availability

Not applicable.

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
