# Peer review of "Crosstalk between Hepatitis B Virus and the 3D Genome Structure"

_viruses, 2022, doi:10.3390/v14020445_

Round 1
Reviewer 1 Report
The manuscript by Dias et al. reviews the current data on the interplay between the Hepatitis B virus and the host genome, with the emphasis on the 3D genome organisation. The work is clear and concise, the references are exhaustive and up-to-date. The MS is overall well-written and provides a brief summary of the field. Nevertheless, I believe the article would benefit from more rigorous definitions of some 3D-genomics terms and additional clarifications of the HBV-related terms (e.g. the names and roles of the HBV proteins mentioned) to aid those using the review to enter the field. Taking this into consideration, I suggest the article be accepted after a minor revision.
Comments:
- Page 3: the authors mention the role of LLPS in the 3D genome organisation and refer to two existing reviews on this topic. For the readers’ convenience, I would suggest including a couple of phrases on LLPS in this very review. At least the concept and the definition of LLPS should be briefly explained. Some explanation/examples of how it may trigger 3D genome organisation would also be useful.
- The authors mention “nuclear bodies”, but only define what they mean by this in the figure capture. As there may be discrepancies regarding this term, it may be better to explain it in the main text. Also, the authors may include the explanations of what are “speckles” and “transcription factories” and how they are recognised (eg. “transcription factories are submicron nuclear regions enriched in RNA polymerase II complexes and associated with nascent RNA production”).
- Page 5: Please, introduce HBx, since it may not be familiar to researchers outside of the HBV field.
- The authors conclude that HBV DNA preferentially contacts the active genome regions. Multiple statements in the review also mention the increased HBV binding on chr19 specifically. The works in refs 52-55 mention a specific region (19p13.11) and suggest the possible mechanism through TF-enhancer interaction. Refs 51 and 56 also mention chr19 as a preferential binding spot. Was the bias towards chr19 also observed in the author's own experimental work referred to in the beginning of the paragraph (page 4)? If so, can the authors provide their understanding of why it is the case and/or describe some features of chr19 making it favourable for HBP binding?
- Page 5, line 19: typo (wrong reference format).
- Page 6, paragraph 2, sentence “Beside the epigenetic state…”. Please, reformulate, the message is vague due to the sentence structure.
- Page 6, paragraph 2, sentence “We do not fully know…”. Could you precise/provide examples of how HBc changes the cccDNA structure?
- The part about the role of HBP integration in cancers (page 8, paragraph 2) is rather independent and may be placed before the conclusions as a separate section entitled accordingly.
- The organisation of the HBV genome (4 promoters, 2 enhancers) is mentioned for the first time in the very last sentence of the article, which does not sound logical. Maybe the authors should move this information to the introduction, and also slightly expand it.
Reviewer 2 Report
In the review “Crosstalk between Hepatitis B virus and the 3D genome structure” by Dias et al, the authors summarize the current knowledge where and when different DNA viruses localize and associate with the cellular genome of infected cells from a 3D perspective. They also describe the different multiple consequences of this association in relation to transcription and DNA replication of the viral as well as host genome.
This short review is well written and recapitulates all major studies and progress, providing the reader a state-of-the-art overview of this fascinating topic. I have a few comments regarding the general layout and structure of the review, which should be addressed by the authors prior to publication:
- “Hepatitis B virus” in the title of the manuscript should be changed to a more general term as the review is not only covering HBV but also other viruses.
- The main part of the review (after the introduction into 3D genome organization) does not contain any subheadings or chapters, making it difficult for the reader to quickly grasp different concepts that are discussed in the review. My suggestion would be to structure this main part a bit more, e.g.
- Interaction and contact sites of viral and cellular genomes
- Functional consequences to the viral and cellular genomes
3) It would be nice if the authors could include a second figure that is summarizing the main aspects of this review such as the preferential genome contacts and the various functional consequences of these contacts. This is the main conceptual message of the review which should be illustrated in addition to Figure 1 which does only give a general overview of eukaryotic 3D genome architecture.
